# How liquids charge the superhydrophobic surfaces

Yuankai Jin [1,2], Siyan Yang[1], Mingzi Sun [3], Shouwei Gao[1], Yaqi Cheng [2], Chenyang Wu[2], Zhenyu Xu[2], Yunting Guo[2], Wanghuai Xu [4], Xuefeng Gao [5], Steven Wang[2], Bolong Huang [3] ✉ & Zuankai Wang [1,6] ✉

Liquid-solid contact electrification (CE) is essential to diverse applications. Exploiting its full implementation requires an in-depth understanding and fine-grained control of charge carriers (electrons and/or ions) during CE. Here, we decouple the electrons and ions during liquid-solid CE by designing binary superhydrophobic surfaces that eliminate liquid and ion residues on the surfaces and simultaneously enable us to regulate surface properties, namely work function, to control electron transfers. We find the existence of a linear relationship between the work function of superhydrophobic surfaces and the as-generated charges in liquids, implying that liquid-solid CE arises from electron transfer due to the work function difference between two contacting surfaces. We also rule out the possibility of ion transfer during CE occurring on superhydrophobic surfaces by proving the absence of ions on superhydrophobic surfaces after contact with ion-enriched acidic, alkaline, and salt liquids. Our findings stand in contrast to existing liquid-solid CE studies, and the new insights learned offer the potential to explore more applications.

Contact electrification is an interfacial process whereby static charges are generated during the contact and separation of two surfaces[1]. CE ubiquitously occurs at various interfaces, particularly at the solid-solid and liquid-solid interfaces. Solid-solid CE has experienced extensive research, and three types of charge carriers[2], including electrons[3], ions[4–6], and materials[7,8], have been used to account for the charge generation during CE between different types of solid surfaces. In comparison, liquid-solid CE is still not well understood despite its significance in various applications, such as water energy harvesting, microfluidics, interfacial chemistry, surface wet cleaning, etc[9–14]. The main bottlenecks hitherto in understanding the mechanism of liquid-solid CE lie in ascertaining charge carriers, which may involve ions, electrons, or both[15–24]. Another challenge in studying liquid-solid CE is achieving quantitative control over the charge generation, which

requires an in-depth understanding of how these charge carriers are dictated by surface properties.

Water has long served as a workhorse in extensive studies of liquid-solid CE due to its ready availability and inherent molecular polarity. It is worth mentioning that charge generation in water or at water-involved interfaces can arise not only from the widely known CE, but also from the other various manners, including introduction and conduction electrification[12,25], as well as the charge transfer across the hydrogen bonds[26–28]. During liquid-solid CE, water is easily positively charged by solid surfaces[29]. Such a characteristic gives rise to the formerly prevalent ion-transfer model, in which anions (hydroxide ions) dissolved in water continually migrate to the solid surfaces due to their high surface affinity, especially those with hydrophobic or superhydrophobic properties, leaving excess positive charges (e.g., hydronium ions) in

[1]Department of Mechanical Engineering, The Hong Kong Polytechnic University, Hong Kong SAR, PR China. [2]Department of Mechanical Engineering, City University of Hong Kong, Hong Kong SAR, PR China. [3]Department of Applied Biology and Chemical Technology, The Hong Kong Polytechnic University, Hong Kong SAR, PR China. [4]Department of Electrical and Electronic Engineering, The Hong Kong Polytechnic University, Hong Kong SAR, PR China. [5]Suzhou Institute of Nano-Tech and Nano-Bionics, Chinese Academy of Sciences, Suzhou, PR China. [6]Research Centre for Nature-Inspired Science and Engineering, The Hong Kong Polytechnic University, Hong Kong SAR, PR China. ✉e-mail: bhuang@polyu.edu.hk; zk.wang@polyu.edu.hk

water[15–17]. Such ion-transfer model also incorporates the consideration of cations affinity toward the solid surfaces to explain the CE between acidic liquids and solid surfaces, denoting that ion propensity on surfaces is dependent on the pH values of liquids[30,31]. However, the ion-transfer model has been highly controversial due to debatable surface affinities of cations and anions in numerous experimental and simulation findings[28,32–36], as well as its deficiency in explaining the recent findings that water can be either positively or negatively charged by a diverse array of solid surfaces (Fig. 1). Later, Wang et al. proposed that both electron and ion transfer are involved in liquid-solid CE, in which electron transfer dominates while ion transfer plays a subsidiary role[18–22]. Nevertheless, such a view fails to elucidate how electron transfer is dictated by surface properties, which is crucial for controlling both the polarity and magnitude of static charges generated in liquid-solid CE. Meanwhile, existing studies also overlook the possibility that transferred ions may come from liquid residues on surfaces[37] since exploited solid surfaces exhibiting hydrophilicity or low hydrophobicity (usually with a contact angle of <120°) inevitably adsorb liquids and ions contained[5], limiting its applicability to the superhydrophobic surfaces that exhibit strong repellence to liquids.

In this work, we present novel insights into liquid-solid CE by designing superhydrophobic surfaces and studying underlying CE mechanisms. We design binary self-assembled monolayer of superhydrophobic surfaces by selecting a pair of mercaptan molecules with opposite electron-donating/accepting propensities, including 1H,1H,2H,2H-perfluorodecanethiol (FDT) and its fluorine-free analog, n-decanethiol (DT). By adjusting the molar ratio of FDT and DT, we are able to modulate the work function of the superhydrophobic surfaces due to the opposite dipole of FDT and DT[38,39]. Such superhydrophobic surfaces with regulable work functions enable unprecedented control of both the polarity and magnitude of static charges generated during CE. We reveal that the superhydrophobic surfaces can either positively or negatively electrify the liquids, filling the unexplored area in the field of liquid-solid CE (Fig. 1). We also find that the magnitude of static charges linearly varies with the work functions of the superhydrophobic surfaces, which indicates that the electron transfer between the contacted liquid and solid surface is driven by their work function differences. Such findings enable us to calculate the work function of water based on the corresponding values of the superhydrophobic surfaces. In addition, liquid-solid CE occurring on superhydrophobic surfaces eliminates the liquid residues on solid surfaces and accompanied ion transfer, standing in contrast to liquid-solid CE occurring on hydrophilic/hydrophobic surfaces, thus establishing a connection between surface wettability and ion transfer during CE.

## Results

### CE between coalescence-induced jumping droplets and super-hydrophobic surfaces

One typical example of liquid-solid CE is that condensed water droplets on cold superhydrophobic surfaces can acquire positive charges upon their coalescence-induced jumping[17]. To demonstrate such a CE process, we prepared two kinds of superhydrophobic surfaces by depositing self-assembled mercaptan monolayers, including 1H,1H,2H,2H-perfluorodecanethiol (FDT) and n-decanethiol (DT), on the nano-structured copper oxide substrates (Fig. 2a). To visually assess the polarity of charges in jumping droplets, we applied an upward electric field (with a strength of ~15 kV m⁻¹) by using the homemade setup illustrated in Fig. 2b. Consistent with the findings in the previous study[17], droplets jumping from the superhydrophobic FDT surfaces acquire positive charges, evidenced by their continuous ascent motion under the upward electric field (Fig. 2c). However, the droplet jumping from superhydrophobic DT surfaces only attains a limited height under the upward electric field (upper part of Fig. 2d), which allows the rational speculation from two possibilities. First, jumping droplets acquire limited or negligible positive charges, leading to insufficient upward electric forces on droplets. Second, jumping droplets acquire negative charges, subjecting them to a downward electric force. To clarify this issue, we reversed the direction of the electric field and observed the continuous ascent motion of condensate droplets jumping from the DT surfaces (Supplementary Fig. 1a), proving the generation of negative charges in condensate droplets during liquid-solid CE (lower part of Fig. 2d). Based on the force balance on uniformly moving droplets (Supplementary Fig. 1b), we calculated the droplet charges ($q$) by using the following equation: $q = (\mathbf{F}_D + mg)/\mathbf{E}$, where $\mathbf{F}_D$ is the air drag force derived from the Stokes' Law, $m$ is the mass of droplet, $\mathbf{E}$ is the strength of applied electric force, respectively. According to the detailed discussion in Supplementary Note 1 and Supplementary Fig. 2, the charge magnitude of a randomly selected jumping droplet (with a radius of ~10.3 μm and a terminal velocity of ~11.9 mm s⁻¹) is calculated as −5.77 fC. We also analyzed the radius distribution of the jumping droplet and corresponding charge magnitude (Supplementary Fig. 3) by employing the methodology reported in previous work[17]. Such results provide the first evidence that water can also acquire negative charges upon contact with superhydrophobic surfaces, filling the unexplored area in the field of liquid-solid CE shown in Fig. 1.

### Simulation for the electronic structures of FDT and DT

To understand why FDT and DT determine the charge polarity during liquid-solid CE, we conducted density functional theory calculations to compare the inherent differences in their electronic structures. The detailed computational setup is described in the Supplementary Note 2. Figure 3a, b show the electronic distributions of FDT and DT, respectively, where the FDT exhibits a highly electron-rich state due to its abundant fluorine atoms while the DT displays a highly electron-deficient state, and bonding orbitals near the Fermi levels of both FDT and DT are located on the sulfur sites. Such electronic distribution indicates that FDT tends to accept the electrons from water during CE, leading to positive charges in water. In contrast, DT prefers to donate electrons to water, leading to negative charges in water. Then, we compared the electronic structures based on the projected partial density of states (PDOSs) (Fig. 3c, d). Notably, S-3*p* orbitals of FDT show a sharp peak on the Fermi level, which facilitates the electron transfer from the surface adsorbed water molecules to the electron reservoir formed by fluorine sites. The good overlapping between *p* orbitals in FDT is also revealed, which guarantees the electron transfer.

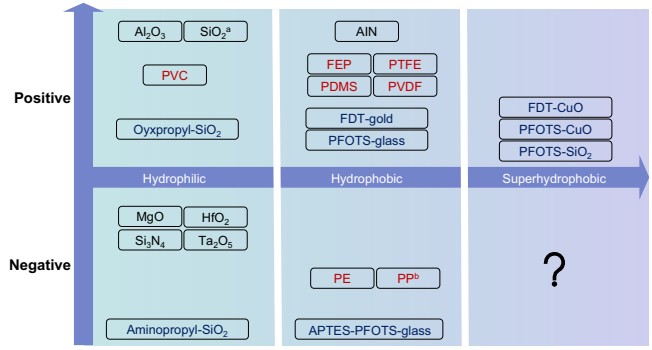

**Fig. 1 | Classification for the existing studies on liquid-solid CE based on the wettability of solid surfaces and charge polarity of water.** There are three types of solid surfaces, including polymer (FEP, etc.)[18,22,51], self-assembled monolayer (PFOTS-SiO₂, etc.)[17,21,52–54], and inorganic solid that is chemically active in water (SiO₂, etc.)[20], which are distinguished by three different font colors. The solid surfaces are located in the position of their corresponding wettability. The positive and negative areas represent the polarity of water charges generated during liquid-solid CE. The sign "?" means the area yet to be explored, *i.e.*, whether the water can acquire the negative charges from the superhydrophobic surfaces. Note that some works report mutually conflicting results regarding the charge polarity or the surface wettability. For example, SiO₂ is reported as hydrophobic in literature[21] (marked by[a]), and water is positively charged by PP in literature[51] (marked by[b]).

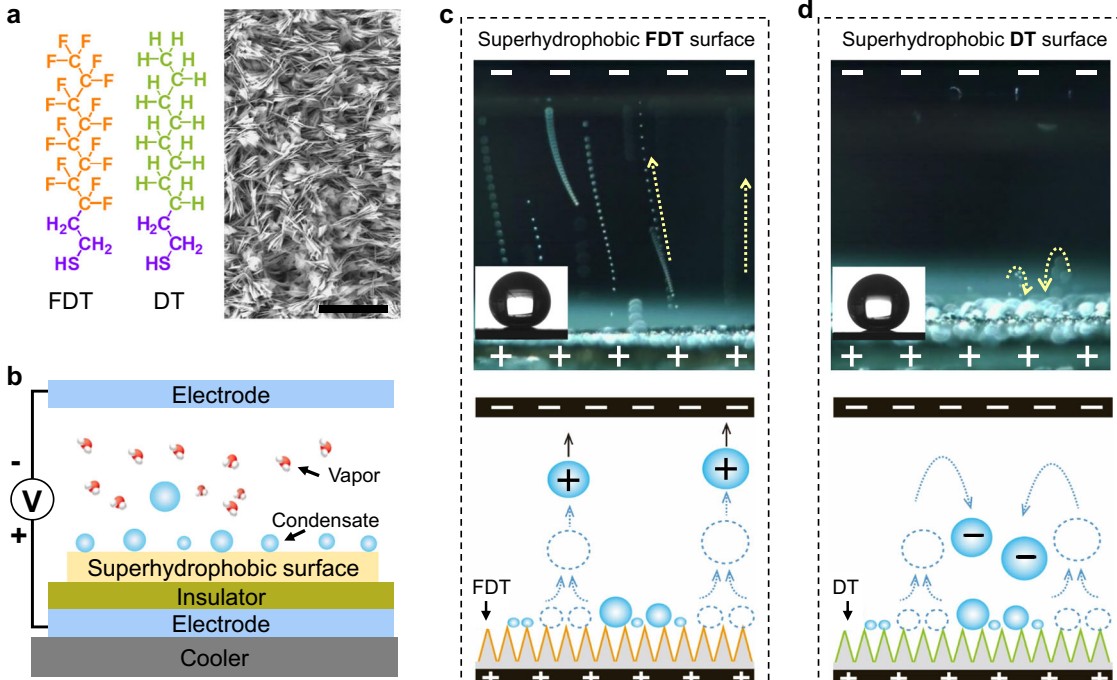

**Fig. 2 | CE between coalescence-induced jumping droplets and super-hydrophobic surfaces. a** Molecular structure of FDT and DT, as well as the morphology of nano-structured copper oxide substrate with 12 minutes of etching. The scale bar is 5 μm. **b** The schematic of the setup that allows the observation of the contact electrification between superhydrophobic surfaces and coalescence-induced jumping of condensate droplets by decreasing the temperature of superhydrophobic surfaces. The cooler aims to trigger the condensation of vapor and coalescence-induced jumping of condensate droplets by decreasing the temperature of superhydrophobic surfaces. Two electrodes build the directional electric field, allowing us to determine the charge polarity of jumping droplets by observing their motion trajectory. **c** On superhydrophobic FDT surfaces, jumping droplets experience a continuous ascent motion under the upward electric field, denoting the generation of positive charges in droplets during CE. **d** On super-hydrophobic DT surfaces, jumping droplets acquire negative charges, leading to initial jumping and followed falling motion under the upward electric field. The contact angles of water droplets on FDT and DT surfaces are ∼ 156.2 ° and ∼ 155.4°, respectively, shown in the insets.

FDT also shows a slightly enlarged bandgap in comparison to DT, meaning a higher barrier for electron depletion. We further compared the *p*-band center of the overall structures and S sites (Fig. 3e). FDT exhibits a much lower *p*-band center than DT, indicating the more electron-rich features induced by the abundant fluorine atoms, supporting their electronic distributions. Meanwhile, the *p*-band center of sulfur sites in FDT is higher than in DT, which demonstrates a higher activity of sulfur sites in FDT.

## CE between water droplets and binary superhydrophobic surfaces with tunable work functions

The simulation results reveal the opposite preference of FDT and DT to accept or donate electrons, which provides a potential means to quantitatively control the charge generation during liquid-solid CE. We fabricated a series of binary superhydrophobic surfaces by blending FDT and DT together, denoted by the feeding molar ratio of FDT, referred to as $k$. And $k = \frac{m_{FDT}}{m_{FDT} + m_{DT}}$, where $m_{FDT}$ and $m_{DT}$ represents the molar concentration of FDT and DT, respectively. Note that $k$ does not directly correspond to the actual ratio of FDT on the surfaces. The as-fabricated binary self-assembled monolayers of superhydrophobic surfaces with varied $k$ values are verified using the X-ray photoelectron spectroscopy spectra of three main elements in FDT and DT molecules, including sulfur, carbon, and fluorine elements (Fig. 4a). With the increase of $k$ values, the sulfur element maintains a consistent binding energy due to the same sulfur bond present in DT and FDT. Meanwhile, the carbon element gradually shifts to a higher binding energy, and the fluorine element shows an intensified peak intensity, indicating a successful modulation of the ratio of FDT and DT on binary superhydrophobic surfaces. As expected above, these super-hydrophobic surfaces with varied $k$ values directly affect both the

magnitude and polarity of charges in coalescence-induced jumping water condensates (Supplementary Note 3 and Supplementary Fig. 4). The maximum negative and positive static charges are generated from the superhydrophobic DT surfaces ($k = 0$) and superhydrophobic FDT surfaces ($k = 1$), respectively, and there is a crossover of charge polarity when $k$ lies between 0.1 and 0.15.

To further understand the influence of $k$ values on CE, we characterized the work functions, the electric properties that reflect the surface capability to donate or accept electrons, of the super-hydrophobic surfaces and measured the static charges in water, respectively. To determine work functions, we first measured the surface potential ($V_{sp}$) of superhydrophobic surfaces, a typical electrical property in characterizing the surface charging ability[40]. Surface potential decreases from ∼ 812 mV to ∼ −1226 mV (Fig. 4b and Supplementary Fig. 5) with the $k$ values increasing from 0 to 1. We then transformed the values of surface potential into work functions by using the equation $\varnothing_{sur} = \varnothing_{tip} - e \cdot V_{sp}$, where $\varnothing_{sur}$, $\varnothing_{tip}$ is the work function of superhydrophobic surfaces and used probes (∼5.02 eV), respectively. Figure 4c denotes the varied work functions of binary superhydrophobic surfaces with the $k$ values. Further, we quantified the droplet charges ($Q$) generated during CE by collecting the water droplets in a Faraday cup that is connected to a Coulomb meter (Fig. 4d). Water droplets acquire static charges upon impacting and rebounding from the superhydrophobic surfaces. We discovered that there is a linear relationship between droplet charges and work functions of superhydrophobic surfaces (Fig. 4e), which is further fitted by a dimensionless equation expressed as $Q = 1.49(\varnothing_{sur} - 5.10)$ with a R-Square value of ∼0.993. Note that the slope of this equation may vary depending on the actual experimental conditions, as the liquid-solid CE is affected by multiple factors[41], such as surface morphology

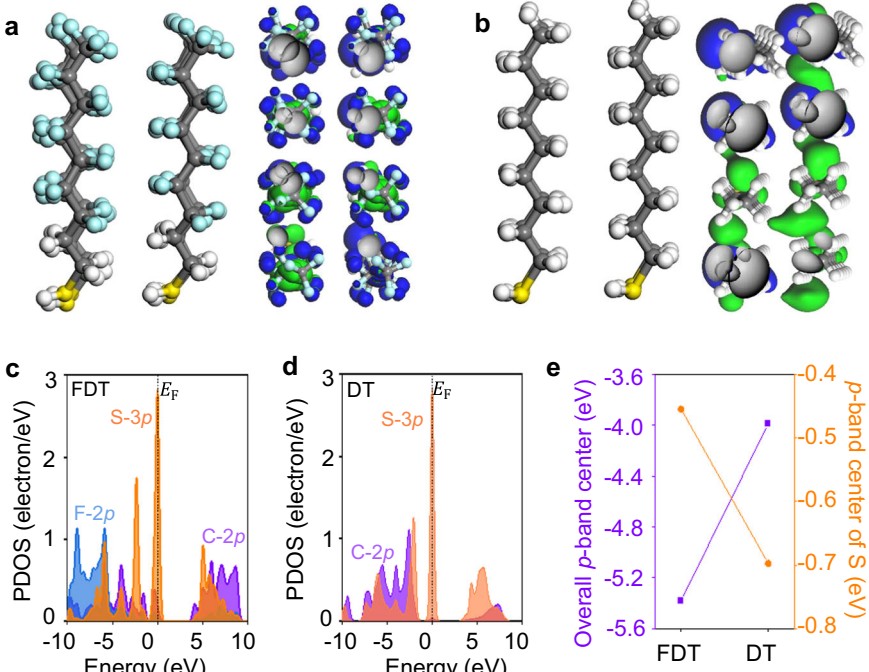

**Fig. 3 | Comparison of FDT and DT from their electronic structures. a, b** The 3D contour plot of electronic distributions of (**a**) FDT and (**b**) DT near the Fermi level. Grey balls = C, Yellow balls = S, Cyan balls = F, and White balls = H. Blue and green isosurface denote bonding and anti-bonding orbitals, respectively. FDT exhibits a highly electron-rich state contributed by the bonding orbitals (blue isosurface), tending to accept electrons. In comparison, DT displays a highly electron-deficient surface with limited contributions from the bonding orbitals. **c, d** The PDOS of (**c**) FDT and (**d**) DT, revealing a larger bandgap of FDT than that of DT. Here, F, S, C, and $E_F$, represent fluorine, sulfur, carbon elements and Fermi level, respectively. **e** Difference between FDT and DT in their $p$-band center. The lower overall $p$-band center (left y-axis) of FDT dictates its electron-rich features, whereas the higher $p$-band center of S (right y-axis) demonstrates a higher activity of S sites in FDT to transfer electrons.

(Supplementary Fig. 6) and droplet impact dynamic (Supplementary Note 4 and Supplementary Fig. 7). This correlation also indicates that the magnitude of charges in water is zero when $\varnothing_{sur}$ equals 5.10 eV, and we defined such a value as the zero-charge point (indicated by the red star in Fig. 4e). The zero-charge point represents the crossover point for the polarity of droplet charges generated during CE. When the $\varnothing_{sur}$ of superhydrophobic surfaces is larger or smaller than the zero-charge point, water will correspondingly acquire the positive or negative charges during CE, respectively.

## Mechanism of CE between liquids and superhydrophobic surfaces

To elucidate the mechanism of CE between water and superhydrophobic surfaces, we revisited the prevailing theory concerning CE between liquids and solid surfaces exhibiting hydrophobicity (with a contact angle of <120°) or hydrophilicity[18–22]. When CE occurs, both electrons and ions serving as charge carriers are transferred to the solid surfaces, with electrons being the dominant carriers (left part of Fig. 5a). Previous studies have presented that ions on solid surfaces typically originate from various sources[42], mainly including ionization reaction on chemically active solid surfaces[20], water hydrolysis at the interface[31], and liquid dewetting leaving behind hydration shell (i.e., the residues of liquids and contained ions)[37]. There is an important yet overlooked fact that ion transfer caused by liquid residues inevitably occurs because the previously exploited surfaces always adsorb the liquids during their contact[5], and these ions are difficult to measure directly, especially on hydrophobic surfaces like PTFE[19]. However, transferred ions result in the surface potential change and deteriorate the charging ability deterioration of the solid surfaces[18,20], which could be used as indicators of the occurrence of ion transfer during CE.

Here, we revealed that ion transfer is not involved in the liquid-solid CE occurring on superhydrophobic surfaces by proving the absence of ions on superhydrophobic surfaces after contact with ion-enriched liquids. The direct evidence is the stable surface potential (or work function) and charging ability of superhydrophobic surfaces. Figure 5b demonstrates that the surface potentials of both superhydrophobic DT and FDT surfaces remain basically unchanged after one minute of soaking in various types of liquids, including water, acid (hydrogen chloride), alkali (sodium hydroxide), and salt (sodium chloride) solutions. It is worth mentioning that such independence of surface potential from ions is inherently analogous to the findings obtained through high-sensitivity surface characterization technologies in a recent advancement of electric field-induced hydrodynamics[43]. Fig. 5c depicts the nearly same magnitude of static charges generated in CE between water droplets and ionic solution-impacted superhydrophobic surfaces, indicating the stable charging ability of superhydrophobic surfaces. Such stable charging ability is further consolidated by the time-involved linearly accumulated liquid charges (Supplementary Note 5, Supplementary Fig. 8). These results reveal that superhydrophobic surfaces effectively prevent ion transfer during liquid-solid CE, which benefits from its strong capacity to repel the remnant of liquids and ions contained. Note that failure of superhydrophobic surfaces under certain conditions, such as overloaded hydrostatic pressure or liquid flooding[44,45], may lead to the occurrence of liquid residues induced ion transfer during CE. Simultaneously, CE between liquids and superhydrophobic surfaces also demonstrates a unique property of pH independence due to the absence of ions on surfaces. Briefly, the superhydrophobic surfaces can either positively or negatively electrify the liquids, regardless of the pH values, which is contrary to the existing findings where acid and alkali aqueous solutions can only be negatively and positively charged during CE, respectively. Such difference is further discussed in Supplementary Note 6, Supplementary Figs. 9 and 10. Additionally, the pH independence of liquid-solid CE occurring on superhydrophobic surface also eliminates the

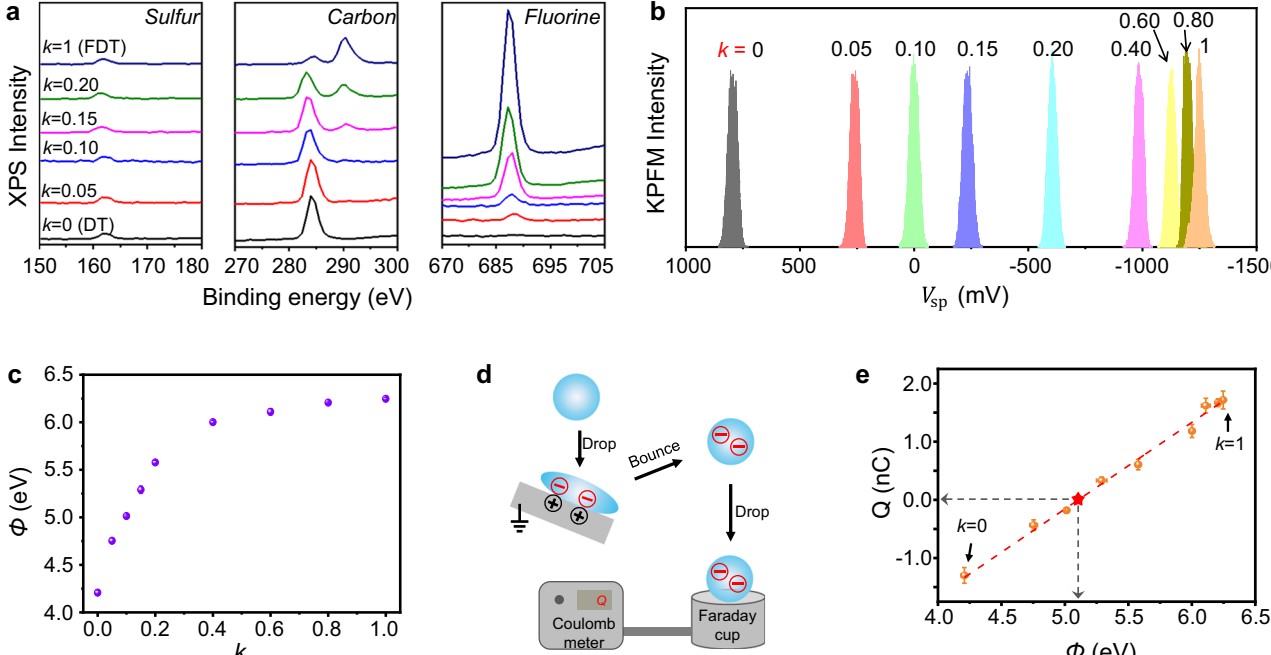

**Fig. 4 | Quantitative regulation for the CE between water and superhydrophobic surfaces. a** XPS results of sulfur, carbon, and fluorine elements on the superhydrophobic surfaces with varied $k$ values (molar ratio of FDT). With the increase of $k$ values, the intensity of fluorine elements gradually increases, and the binding energy of carbon elements gradually shifts to higher values. **b** Surface potential ($V_{sp}$) on the superhydrophobic surfaces with the varied $k$ values, measured using the KPFM. **c** Work functions ($\varnothing$) of the superhydrophobic surfaces with varied $k$ values. **d** Schematic diagram illustrating the measurement of droplet charges ($Q$) generated during CE. Water droplets acquire static charges upon impact and rebound from the grounded superhydrophobic surfaces. Both the magnitude and polarity of droplet charges can be detected by the Faraday cup that is connected to the nanocoulomb meter. **e** The linear correlation between droplet charges and work functions of superhydrophobic surfaces. There is a crossover point (denoted by the red star) for the polarity of charges in water droplets, which is defined as the zero-charge point of water. The error bars in (**c, e**) are based on the SD values of three tests, and the error bars in (**c**) are too small to allow clear differentiation.

potential influence of surface impurities that usually used to account for some phenomena beyond the ion adsorption, given that the surface impurities can induce the pH dependence of charged surface[46,47].

After ruling out the efficacy of ions, electrons emerge as the exclusive carriers for charge generation during CE between liquids and superhydrophobic surfaces (right part of Fig. 5a). In fact, the electron transfer during liquid-solid CE has been well-documented[18–23]; however, it is still unclear how electron transfer is dictated by surface properties. To further understand the electron transfer during liquid-solid CE, we referred to the metal-involved CE because of their similarity manifested in the linear correlation between generated charges and work functions[48]. There has been a consensus that metal-involved CE results from the electron transfer driven by the difference in work functions between the contacted surfaces[1,5]. Based on this understanding, we considered the electron transfer between liquids and superhydrophobic surfaces regarding their work functions. Both FDT and DT have been found to effectively regulate surface work functions by modifying the substrates (copper oxide with a measured work function of ~4.9 eV), which benefit from their dipoles induced by atoms with unequal electronegativity[38,39]. For FDT with trifluoromethyl group, its electric dipole increases the surface work function by shifting the vacuum energy level, $E_{VAC}$ (Fig. 5d, Supplementary Fig. 11a). Conversely, DT with methyl group shows a dipole orientation opposite to that of FDT, leading to the decrease of surface work function (Fig. 5d, Supplementary Fig. 11b). Due to different work functions between surfaces, FDT surfaces acquire the electrons (Fig. 5e) while DT surfaces tend to lose the electrons (Fig. 5f) upon contact with the liquids. Accordingly, the binary superhydrophobic surfaces with varied $k$ values encompassing a range of work functions from $\varnothing_{FDT}$ to $\varnothing_{DT}$ enables the control over electron transfer during liquid-solid CE. Note that when the work function of liquids is exactly matches that of the superhydrophobic surfaces (defined as $\varnothing_0$), electron transfer is prohibited (Supplementary Fig. 11c), resulting in the zero-charge point of liquids (water Fig. 4e). Therefore, the work function of water is calculated as ~5.10 eV.

## Discussion

We design the binary self-assembled monolayer of superhydrophobic surfaces that possess regulable work functions and liquid repellence simultaneously. These two characteristics decouple the electrons and ions due to their respective functions in controlling electron transfer and excluding ion transfer during liquid-solid CE, leading to the intriguing phenomena of liquid-solid CE, as shown below. (1) There is a linear relationship between the work function of superhydrophobic surfaces and generated charges, which allows us to achieve, for the first time, quantitative control over the magnitude of static charges generated during liquid-solid CE. (2) The superhydrophobic surfaces can either positively or negatively electrify the water, salt, acid, and alkali aqueous solutions without deterioration in charging ability, denoting the independence of CE between liquids and superhydrophobic surfaces on the ion types and pH levels of liquids. In contrast, existing liquid-solid CE is subject to the pH values of liquids. A typical example is that the alkali solutions can only acquire negative charges from hydrophilic/hydrophobic surfaces during CE due to the continuous adsorption of OH⁻ on the solid surface, and such ion adsorption will deteriorate the charging ability of the solid surfaces.

In summary, we describe, quantify, and modulate the contact electrification between liquids and superhydrophobic surfaces, thereby revisiting the dynamics of liquid-solid CE. By integrating experimental findings and simulation results, we find that electron transfer, driven by the work function difference between the liquids and solid surfaces, underlies the liquid-solid CE. In addition, CE

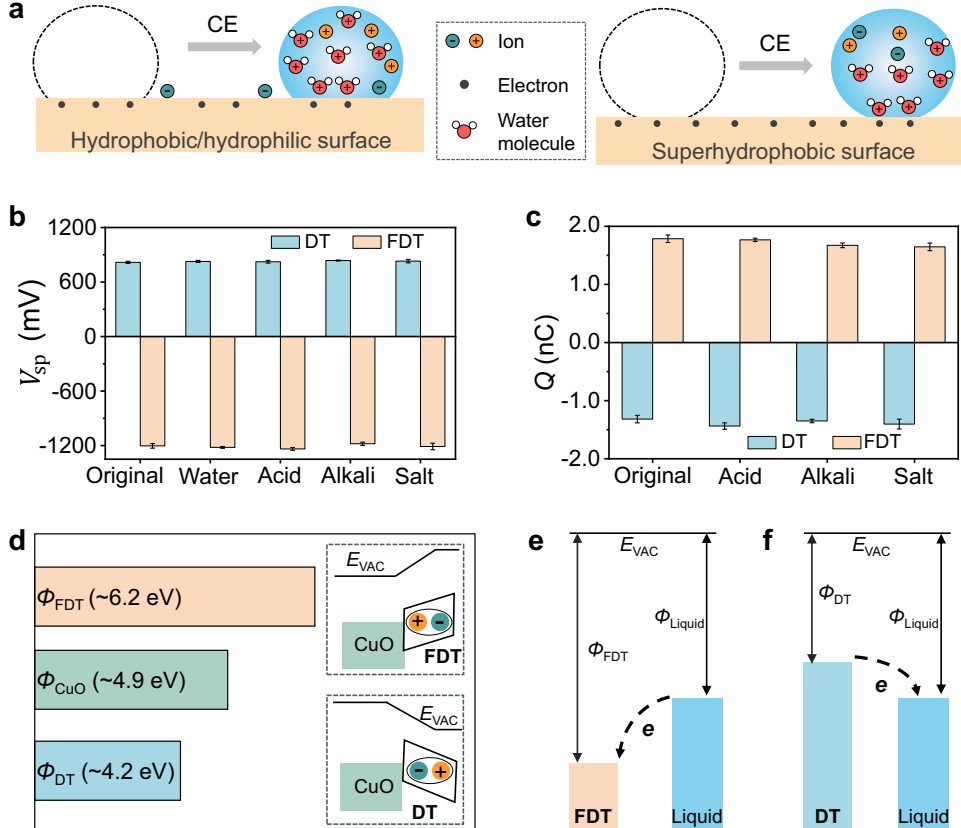

**Fig. 5 | Mechanism of CE between liquids and superhydrophobic surfaces.**
**a** Comparison of liquid-solid CE occurring on hydrophobic/hydrophilic surfaces (left part) and on superhydrophobic surfaces (right part). On hydrophobic/hydrophilic surfaces, both ion and electron transfer occur[18–22], whereas on superhydrophobic surfaces, only electrons remain after the liquid recedes. **b**, **c** The experimental evidence demonstrating the repellency of superhydrophobic surfaces to the ions in liquids. Here, the concentration of the ionic solutions is 0.1 mM, meaning that the pH values of acid and alkali solution are 4 and 10, respectively. **b** Measured surface potentials ($V_{sp}$) of the superhydrophobic surfaces in the original state and after being soaked by the ionic solutions, suggesting stable work functions of superhydrophobic surfaces. **c** Static charges (in water droplets, $Q$) generated in liquid-solid CE occurring on the superhydrophobic surfaces in the original state and after being impacted by the ionic solutions. The results denote

that superhydrophobic surfaces maintain a stable charging ability after experiencing the impact of ionic aqueous solutions. The error bars are based on the SD values of three tests. **d** Measured work function values ($\varnothing$) of copper oxide (CuO), superhydrophobic FDT and DT surfaces. Inserts are the sketch for the capability of FDT and DT to regulate work functions. The interfacial dipole of FDT and DT induces the vacuum energy level ($E_{VAC}$) shift and thereby change the work functions of surfaces (a detailed version is shown in Supplementary Fig. 11a, b). Because dipole moment always points from negative to positive charge, FDT increases the surface work function to ~ 6.2 eV, whereas DT decreases that to ~ 4.2 eV. **e**, **f** Diagram for electron transfer driven by work function difference between the superhydrophobic surfaces and liquids. The electron consistently transfers from the surface with a lower work function to one with a higher work function, that is, from liquid to FDT (**e**), and from DT to liquid (**f**).

occurring on superhydrophobic surfaces excludes the ion transfer process, in contrast to the existing liquid-solid CE occurring on surfaces exhibiting hydrophilicity or low hydrophobicity, which proves that ion transfer during liquid-solid CE is affected by surface wettability. These findings advance the understanding of liquid-solid CE and open potential avenues for further exploration of practical applications. Furthermore, superhydrophobicity-induced unique results on liquid-solid CE are expected to promote the revaluation of previous findings in liquid-solid interfacial research, for example, interfacial chemistry under the situation of liquid flowing along a solid surface[49].

## Methods
### Fabrication of superhydrophobic surfaces with tunable work functions
To obtain the superhydrophobic surfaces, we first chemically etch the cleaned coppers by using a hot (96 °C) solution with a component of NaClO$_2$, NaOH, Na$_3$PO$_4$·12H$_2$O, and H$_2$O at a 3.75:5:10:100 wt. ratio for 12 minutes, forming superhydrophilic copper oxide surfaces with nano-grass structures. Subsequently, the nano-structured surfaces are immersed in a 5 mM alcohol solution of FDT and DT with designed $k$ values for 90 minutes. To dramatically eliminate the potential physical

adsorption, the fabricated superhydrophobic surfaces are carefully rinsed with alcohol and subsequently dried using a nitrogen blow.

### Characterization for the superhydrophobic surfaces
The static contact angles of water on FDT and DT surfaces were measured by a Kruss DSA100 contact angle goniometer at room temperature. Surface elements and their chemical shifts of the superhydrophobic surfaces with varied $k$ values were analyzed by X-ray Photoelectron Spectroscopy, XPS (ESCALAB 250Xi multifunctional spectrometer, Thermo Fisher). The surface potential was measured using Kelvin probe force microscopy (KPFM, Dimension Icon, Bruker). Pt-Ir-coated probe (SCM-PIT-V2, Bruker) was used, and three regions (each with an area of 2 x 2 μm²) were randomly selected from the samples with a size of 1 x 1 cm². A highly oriented pyrolytic graphite (with a work function of 4.4 eV in air[50]) is selected as a benchmark for the work function of superhydrophobic surfaces, and its KPFM image is shown in Supplementary Fig. 12.

### Characterization for the static charges in liquid droplets
The droplet charges, including polarity and magnitude, are detected after the droplets are collected in a Faraday cup connected to a

nanocoulomb meter with a resolution of 0.01 nC (Monroe, Model 284). Since the static charges in a single water droplet are hard to distinguish the difference among the surfaces with varied $k$ values, we recorded the charges in droplets for a duration of 30 s under dropping height of 3 cm and dropping frequency of 6.7 Hz (calculated from the video captured by a high-speed camera, Fastcam SA4, Photron Limited). The volume of an individual water droplet is ~15 μL, which is determined from water density and mass measured using the electronic scale with an absolute accuracy of one ten-thousandth. To mitigate measurement discrepancies, we conducted a thrice-repeated weighing of ten individual droplets and then calculated the average values.

### The experiments for proving the absence of ions on the super-hydrophobic surfaces

Three types of ionic solutions, including sodium chloride, hydrogen chloride, and sodium hydroxide, were used to soak or impact the superhydrophobic FDT and DT surfaces. In the soaking experiments, the surface potentials of freshly fabricated superhydrophobic FDT and DT surfaces were first measured. The surfaces were then soaked in the target ionic solutions, such as hydrogen chloride solution, for 1 minute. The superhydrophobic nature of surfaces prevents the water residual, allowing the followed direct measurements of surface potential again without any additional treatment. Such a soaking treatment and surface potential measurement were repeated twice, in which the other two kinds of ionic solutions were respectively used to soak the surfaces. As for the impacting experiments, the charges in water droplets generated from CE with FDT and DT surfaces were first measured using the methods described above. Subsequently, the superhydrophobic surfaces were subjected to the impact of the ionic solution for 1 minute. Then, the treated surfaces were impacted by the water droplets while simultaneously recording the charges in droplets. Such droplet impacts and charge measurements were repeated twice, and two other kinds of ionic solutions were used to impact the surfaces on each occasion. Here, the impact points on the surface by water and ionic solution droplets are the same. Note that the measured results are independent of the utilization orders of ionic solutions.

## Data availability

The authors declare that the data supporting the findings of this study are available within the paper and its supplementary information files. Source data are provided with this paper.

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

## Acknowledgements

We acknowledge the financial support from the National Natural Science Foundation of China (No. T2293694, 51975502), Research Centre for Nature-inspired Science and Engineering. M.S. and B.H. were supported by the Research Centre for Carbon-Strategic Catalysis and Projects of Strategic Importance of the Hong Kong Polytechnic University (1-ZE2V).

## Author contributions

Y.J. and Z.W. conceived the research. Z.W. and B.H. supervised the research. Y.J., S.Y., S.G., Y.C., C.W., Y.G., Z.X., and W.X. carried out the experiments. B.H. and M.S. conducted the simulation. S.W. and X.G. provided support in terms of sample preparation. All authors analyzed the data. Y.J., Z.W., and B.H. wrote the manuscript, and all the authors agreed on its final contents.

## Competing interests

The authors declare no competing interests.
