## [Peer Review File · Nature Communications]

Reviewers' comments:

Reviewer #1 (Remarks to the Author):

In this joint experimental and theory work, Wang and co-workers tackle a very interesting and challenging problem related to the origin of charging

at hydrophobic surfaces. This is a topic that is extremely interesting and whose underlying physics and chemistry remains poorly understood.

I think that the work certainly has potential for impact and to contribute to the ongoing discussion in the field of contact electrification. There are however some important points that the manuscript misses/does not clarify regarding their assumptions on the mechanisms - this also has important implications on how they interpret their results. The authors would need to probably conduct new experiments and calculations to validate/reinforce their findings.

I list the important ones here:

1) One key assumption/ flaw of the paper is that the authors have missed an extensive amount of literature showing that charging can arise from very different sources - namely charge transfer from water to hydrophobic surfaces. This was first suggested by Roke and co-workers in a joint theory and experimental paper and then later by Poli and Hassanali. It was then confirmed by surface sensitive experiments. All the relevant papers are indicated below:

<https://pubs.acs.org/doi/10.1021/ja202081x>

<https://www.nature.com/articles/s41467-020-14659-5>

<https://www.science.org/doi/10.1126/science.abj3007>

2) In addition, another important point to pay attention to is that there are several theoretical studies showing that ions like OH⁻ are actually repelled from the interface. The authors should take these studies into account in their interpretation of their results:

<https://pubs.acs.org/doi/10.1021/jacs.5b07232>

3) If water can acquire negative charges from surfaces (in other words, excess electrons), then these excess electrons should be probed by spectroscopy - the UV absorption of the excess electron has a very distinct signature and could be probed experimentally. I understand that these are difficult experiments to do but I think there needs to be more proof that this mechanism is actually operational. It would also significantly enhance the impact of the paper.

4) In Figure 5B the authors say that they do not find a change in the surface potential going from water to HCl to NaCl implying there is no specific binding of the ions. The authors may want to look at the following paper by the Roke group where very similar findings are unravelled using surface-sensitive techniques:

<https://chemrxiv.org/engage/chemrxiv/article-details/6493fdeb853d501c004fb46d>

5) I am extremely concerned with the quality of the electronic structure

calculations using a standard GGA functional PBE. This is well known to over-delocalize charge and lead to spurious charge-transfer effects. The authors should repeat and validate their calculations with either hybrid functionals OR with using DFT+U (Hubbard Corrections) OR with scissor corrections all of which are schemes to reduce the error of the self interaction. Without these benchmarks and tests, the results cannot be trusted.

Reviewer #2 (Remarks to the Author):

Jin et al. analyze water drops, condensing on superhydrophobic surfaces. It had been observed (Miljkovic et al., 2013, ref. 17) that after nucleation and growth, water drops often jump up after merging and carry a positive charge. This fact has let a number of studies where this charge was used to enhance condensation. Jin et al. use a specific material, namely Cu coated with nanoneedles, to condense droplets. They coat their surfaces with a mixture of alkanethiols and fluorinated alkanethiols. Positively charged drops were observed when the drops jumped from fluorinated alkanethiols. Results indicate that the alkanethiol coating leads to negatively charge jumping droplets. It is the first time that negatively charge drops are observed on superhydrophobic surfaces; on hydrophobic surfaces negatively charge drops have been observed before. From a comparison with density functional theory calculations and Kelvin potential measurements the authors conclude that on the superhydrophobic surfaces charging is due to electron transfer.

In my opinion the results are relevant to a specialized audience. The experimental part should be described more in detail. I find the arguments that electrons rather than ions are transferred not convincing. For this reason, I do not recommend to publish in Nature Communications.

An important part is the determination of the drop charges. To obtain drop charges, the authors analyze the trajectory of jumping droplets. How was the size of the droplets measured? I guess from high speed videos. The radius is reported with high precision (10.3 μm). What is the error? Pictures of the droplets should be provided as examples to demonstrate that drop radii could be measured precisely. I guess that jumping drops were not all of precisely the same size. What was the distribution? Since the drop mass depends critically on the droplet radius, it would be good to estimate the error in drop mass. How many drops were analyzed?

The Copper nanograss is conducting. When applying an electric field, the field strength at the top of the needles may be very high. How can the authors exclude that such a high field strength does not change the drop charge? Electrons may be ejected due to the sharp ends of the fibres.

Such a surface with needles may also be critical for the Kelvin potential measurements. The surface is certainly not homogeneous and extremely rough. Which part of the surface is dominating the measured Kelvin potential? How does the Kelvin potential vary across the surface? Is gold a good reference for KPFM? It is notoriously contaminated.

A figure of the setup should be in the main manuscript. Now it is in SI as figure S1. I also suggest adding an SEM image of the Cu nanograss.

Reporting the density of water with 6 digits precision (note S1) does not make much sense. Especially because the temperature is not reported.

The authors argue that on hydrophobic surfaces electrons and ions remain on the surface while on superhydrophobic surfaces only electrons remain (e.g. figure 5 caption). What is the evidence? I do not see convincing arguments. The soaking experiments are not convincing to me. It is not clear at all, what the soaking does to the thiol layer on the oxidized copper.

Reviewer #3 (Remarks to the Author):

The authors investigated the mechanisms governing surface-droplet contact electrification using superhydrophobic surfaces that decouple ion transfer and electron transfer. The manuscript was well written and the experiments were well performed. I also like the idea of using binary self-assembled monolayer for a delicate control of droplet charging. In order to further reinforce the strength of the study, the following aspects should be addressed with additional clarity:

1. The title and Figure 1 appear to emphasize the unique mechanism of contact electrification associated with superhydrophobic surfaces in comparison to hydrophilic and hydrophobic surfaces. Based on my understanding, superhydrophobic surfaces can be regarded as an extension of the hydrophobic case when considering droplet-surface interactions at a microscale level. It remains unclear what fundamentally differentiates droplet-contact induced electrification on smooth hydrophobic surfaces from superhydrophobic surfaces. If the distinction lies in the presence of ion residue, it would be valuable to discuss the implications of using a slippery surface.

2. The claim that superhydrophobic surfaces eliminate liquid and ion residues should be approached with greater rigor. From a microscopic perspective, the droplet interacts with the surface structures,

and it is plausible that residues persist on the micro/nanoroughness after liquid-droplet separation. This is especially relevant when the droplet possesses a higher temperature than the substrate, as condensation underneath the droplet (for instance, ACS nano, 13(7), 8169-8184, 2019) during impact may result in the retention of residues on the surface.

3. A more comprehensive characterization of the binary self-assembled monolayers employed to achieve superhydrophobicity is suggested, particularly with regard to the uniformity and thickness of the coating. In addition to chemical adsorption, physical adsorption can occur in certain cases, potentially resulting in the deposition of additional layers of molecules onto the monolayer. This phenomenon may surpass the controllability achieved solely by manipulating the molar ratio of molecules.

4. Although the driving force (work function) has been identified, the impact of droplet-surface interaction on electron transfer remains to be further illuminated. The authors have demonstrated that the obtained charges are also influenced by the surface morphologies of the superhydrophobic surfaces. This suggests that droplet impact dynamics may affect the actual contact area between the droplet and the surface, consequently influencing the quantity of charges acquired. Bridging the gap between droplet charges and droplet-surface interaction would significantly enhance the current study.

5. The authors primarily quantified the charge of the droplet rather than that of the surface. Therefore, "How do the superhydrophobic surfaces charge the liquid droplet?" seems a more suitable title than the current version, "How does the liquid charge the superhydrophobic surface?"

6. Lastly, the following typos should be corrected:

- "Self-assembly monolayer" should be written as "self-assembled monolayer."

- In Supplementary Note S1, the sign of the gravitational force mg should be positive.

Reviewer #1

In this joint experimental and theory work, Wang and co-workers tackle a very interesting and challenging problem related to the origin of charging at hydrophobic surfaces. This is a topic that is extremely interesting and whose underlying physics and chemistry remains poorly understood.

I think that the work certainly has potential for impact and to contribute to the ongoing discussion in the field of contact electrification. There are however some important points that the manuscript misses/does not clarify regarding their assumptions on the mechanisms - this also has important implications on how they interpret their results. The authors would need to probably conduct new experiments and calculations to validate/reinforce their findings.

I list the important ones here:

Response: We appreciate the reviewer for recognizing the importance of our work. We also thank the reviewer for the professional suggestions to enhance the research background, experiments, and calculations of this work.

1. One key assumption/flaw of the paper is that the authors have missed an extensive amount of literature showing that charging can arise from very different sources - namely charge transfer from water to hydrophobic surfaces. This was first suggested by Roke and co-workers in a joint theory and experimental paper and then later by Poli and Hassanali. It was then confirmed by surface sensitive experiments. All the relevant papers are indicated below:

<https://pubs.acs.org/doi/10.1021/ja202081x>

<https://www.nature.com/articles/s41467-020-14659-5>

<https://www.science.org/doi/10.1126/science.abj3007>

Response: We greatly appreciate these critical references that help us gain a better understanding of how water acquires charges on a microscopic level. In fact, the second reference by Poli was cited in our previous version of the manuscript but in an improper position. After carefully referencing the works by the Roke group, we are able to effectively build the connection between the charging manners of water across the hydrogen bonds and the methods we primarily focused on before, which is crucial for us and this manuscript. We have cited these references as Ref. 26-28.

2. In addition, another important point to pay attention to is that there are several theoretical studies showing that ions like OH⁻ are actually repelled from the interface. The authors should take these studies into account in their interpretation of their results:

<https://pubs.acs.org/doi/10.1021/jacs.5b07232>

Response: We thank the reviewer again for providing us with this valuable reference. The reference theoretically reveals that hydroxide ions (OH⁻) are repelled while the protons are attracted to the surface, showing a contradictory propensity with the studies in explaining the ion-transfer model of liquid-solid contact electrification (J. Phys. Chem. B 2001, 105, 8544; J. Am. Chem. Soc. 2008, 130, 3915). Such a contrasting view is attributed to the utilization of different characterization methods, which is also mentioned in the JACS paper by Roke group, the reference suggested by the reviewer in comment 1. We have emphasized the issues concerning the surface affinity of hydroxide ions in the revised manuscript (Line 59 Page 3)

Regarding the effect of ion surface affinity on our results, the liquid-solid contact electrification occurring on superhydrophobic surfaces demonstrates independence on the ions (including cations and anions) and their surface affinity because the superhydrophobic surfaces can either positively or negatively triboelectrified the acid, alkali, and salt solutions regardless of types of cations and anions in the liquid.

3. If water can acquire negative charges from surfaces (in other words, excess electrons), then these excess electrons should be probed by spectroscopy - the UV absorption of the excess electron has a very distinct signature and could be probed experimentally. I understand that these are difficult experiments to do but I think there needs to be more proof that this mechanism is actually operational. It would also significantly enhance the impact of the paper.

Response: We greatly appreciate this advice on characterizing the transferred electrons using spectroscopy. However, the transferred electrons during liquid-solid contact electrification may transform into other kinds of species, such as hydroxyl radicals, as documented in a recent work (Nano Energy, 2023, 112: 108464). Consequently, it is challenging to deduce the existence of electrons solely through spectroscopy without further in-depth research, which we hope to conduct in future work.

Additionally, the electron transfer during liquid-solid has been extensively proven based on electron properties in thermal emission (Nat. Commun. 2020, 11, 399) and chemical activity (Nature Communications, 2022, 13(1), 130). Nevertheless, these studies fail to elucidate how electron transfer is dictated by surface properties. Building upon this foundation, we further resort to the binary superhydrophobic surfaces with adjustable surface potentials to elucidate the relationship between the charges generated and the surface work function, thereby proposing that electron transfer during liquid-solid contact electrification is driven by the work function difference of two surfaces.

This comment reminds us to strengthen the claim regarding electron transfer during liquid-solid contact electrification further. Accordingly, we have included supplementary claims in the revised manuscript (Line 207, Page 9)

4. In Figure 5B the authors say that they do not find a change in the surface potential going from water to HCL to NaCl implying there is no specific binding of the ions. The authors may want to look at the following paper by the Roke group where very similar findings are unraveled using surface-sensitive techniques: <https://chemrxiv.org/engage/chemrxiv/article-details/6493fdeb853d501c004fb46d>

Response: We have carefully read this paper that challenges the conventional understanding of the driving force for electric field-induced hydrodynamics. The authors employ a combination of surface-sensitive techniques and quantum-level computations to provide compelling evidence. A key supporting argument from this reference is that the surface potential of used nanodroplets remains unchanged with the ions (or pH) due to the absence of ions binding, which is similar to the situation of contact between liquids and superhydrophobic surfaces in our work. We believe that drawing parallels from distinct focuses could spur more innovation and discoveries in surface science; therefore, we have introduced this work in the revised main text (Ref. 38).

5. I am extremely concerned with the quality of the electronic structure calculations using a standard GGA functional PBE. This is well known to over-delocalize charge and lead to spurious charge-transfer effects. The authors should repeat and validate their calculations with either hybrid functionals OR with using DFT+U (Hubbard Corrections) OR with scissor corrections all of which are schemes to reduce the error of the self-interaction. Without these benchmarks and tests, the results cannot be

trusted.

Response: We highly appreciate reviewer's kind suggestions on the functional selections of theoretical calculations. DFT based on GGA functionals has achieved great success in the efficiency and accuracy of electronic structure calculations and has become the most popular calculation scheme in multielectronic systems. In this work, all the involved elements in the theoretical calculations are the main group non-metal elements, including F, S, and P, which are composed of only *s* and *p* orbitals. The applications of DFT calculations with GGA and PBE functionals can describe most elements with *s* and *p* orbitals very well, except for those elements with strongly localized and correlated valence electrons, which usually occurs for special material systems such as transition-metal oxides, rare earth metal elements, and rare earth compounds. This is due to the existence of *d* or *f* electrons, which belong to a strongly correlated electronic system and the exchange interaction between electrons cannot be ignored. The Hubbard U correction method can describe the electronic structure with a strong d-electron or f-electron system by adjusting the U parameter, especially for the band gap of transition-metal oxide semiconductors. However, since our calculated systems do not have strong localized electrons induced by *d* or *f* orbitals, we think that our calculations based on GGA+PBE functionals are fully reliable to demonstrate the electronic structures of the material systems in this work.

Reviewer #2

Jin et al. analyze water drops, condensing on superhydrophobic surfaces. It had been observed (Miljkovic et al., 2013, ref. 17) that after nucleation and growth, water drops often jump up after merging and carry a positive charge. This fact has let a number of studies where this charge was used to enhance condensation. Jin et al. use a specific material, namely Cu coated with nanoneedles, to condense droplets. They coat their surfaces with a mixture of alkanethiols and fluorinated alkanethiols. Positively charged drops were observed when the drops jumped from fluorinated alkanethiols. Results indicate that the alkanethiol coating leads to negatively charge jumping droplets. It is the first time that negatively charge drops are observed on superhydrophobic surfaces; on hydrophobic surfaces negatively charge drops have been observed before. From a comparison with density functional theory calculations and Kelvin potential measurements the authors conclude that on the superhydrophobic surfaces charging is due to electron transfer.

In my opinion the results are relevant to a specialized audience. The experimental part should be described more in detail. I find the arguments that electrons rather than ions are transferred not convincing. For this reason, I do not recommend to publish in Nature Communications.

Response: Many thanks for the reviewer's critical comments and suggestions, which greatly help us strengthen this work.

Reviewer's comments mainly focus on three points, including target audience, experimental details, and mechanism of this work. We have addressed each of these points individually, along with our corresponding revisions in the manuscript. We hope that our response and revised manuscript meet the publication standards of Nature Communications.

1. Concerning the target audience, we respectfully disagree with the reviewer. Reviewer notes that the work (by Miljkovic et al.) has led to various studies in the field of heat and mass transfer. Our work reevaluates the underlying mechanism of Miljkovic's work, which would also attract the attention of researchers in the same field in our views. Additionally, our work establishes a crucial connection between contact electrification and wettability, which are subjects of interest to a broad scientific readership engaged in heat and mass transfer, interfacial chemistry and physics, surface

engineering, as well as fluid dynamics.

2. According to the reviewer's comments on experiment details, especially about droplet volume characterization and surface potential measurement, we have provided a detailed response in the following text and have incorporated pertinent revisions into the main text and supporting information. We have also considered experimental details raised by other reviewers in the revised submission. We are open to further suggestions for improvement in this regard.
3. The reviewer expresses concern about the mechanism of this work and the potential lack of novelty. We clarify our key viewpoints in the response to Comment 6 to address the concerns regarding mechanisms. Additionally, it is imperative to highlight that the generation of negative droplet charges during contact electrification (CE) occurring on superhydrophobic surfaces is one facet distinguishing our study from previous research. Our study also unveils other unique experimental phenomena and mechanisms, involving the linear correlation between charges and surface property, distinct ion adsorption behaviors, and pH independence. These findings suggest that our study of CE between liquid and superhydrophobic surfaces is not a simple extension of prior studies at the aspect of surface wettability or charge polarity.

1.1 An important part is the determination of the drop charges. To obtain drop charges, the authors analyze the trajectory of jumping droplets. How was the size of the droplets measured? I guess from high speed videos. The radius is reported with high precision (10.3 μm). What is the error? Pictures of the droplets should be provided as examples to demonstrate that drop radii could be measured precisely. I guess that jumping drops were not all of precisely the same size. What was the distribution?

Response: We concur with the importance of droplet charges and sizes. Our study encompasses an analysis for the charges of jumping droplets (Figure 2) and charges of droplets impacting superhydrophobic surfaces (Figure 4). This comment mainly focuses on the former.

The jumping droplets serve to qualitatively demonstrate the polarity of generated charges during CE, and their charge calculations provide additional evidence for this phenomenon. Considering the broad size and charge distribution of jumping droplets, as referred by this comment, we calculated the charges of one droplet that is randomly selected from countless jumping droplets, and the radius of such a

droplet is ~ 10.3 μm . This point has been articulated more precisely to preclude any misinterpretation in the revised manuscript (Line 108 Page 5) and supporting information.

To calculate the charges of jumping droplets, we analyzed the jumping velocity and size of droplets using the high-speed camera. To be honest, the acquired droplet size may not be exceedingly precise due to the variable alignment with the camera's focal plane, but such a result, we think, offers solid evidence for the generation of negative droplet charges. A more accurate analysis of droplet size and charges is presented in the response to comment 1.2 and Figure 4 of the manuscript.

Motivated by this comment, we have also analyzed both the charge and size distribution of droplets jumping from superhydrophobic DT surfaces (Figure R1, also see Figure S3), employing the methodology described by Miljkovic et al. (Nat. Commun. 2013, 4, 2517).

Figure R1. Radius distribution of droplets jumping from superhydrophobic DT surfaces and corresponding magnitude of negative charges in droplets.

1.3 Since the drop mass depends critically on the droplet radius, it would be good to estimate the error in drop mass. How many drops were analyzed?

Response: We stand with the reviewer's views on the precise characterization of droplet radius. In the part of the quantitative analysis of droplet charges (Figure 4 in the main text), we determined the

droplet size by measuring its mass. The droplet mass is gauged using the electronic scale with an absolute accuracy of one ten-thousandth. To mitigate measurement discrepancies, we conducted a thrice-repeated weighing of ten individual droplets and then calculated the average size of single droplet (~15 μ L). We have added this important experimental detail in the revised manuscript (Line 278 Page 11).

2. The Copper nanograss is conducting. When applying an electric field, the field strength at the top of the needles may be very high. How can the authors exclude that such a high field strength does not change the drop charge? Electrons may be ejected due to the sharp ends of the fibres.

Response: This is a critical comment. Figures R2A, B (also Figures 1C, D) illustrate that, under the electric field directed upward, droplets acquire positive charges and negative charges from superhydrophobic FDT and DT surfaces, respectively. The negative charges in droplets jumping from superhydrophobic DT surfaces are further corroborated in Figure R2C (also see Figure S1A). These results suggest that the electric field is not responsible for the charge polarity. Such results benefit from our designs incorporating an insulator (insulative tape, as shown in Figure 1 of the revised version) between the superhydrophobic surfaces and the electrodes, thereby eliminating the influence of the electric field on droplet charges.

Figure R2. The motion of the droplets jumping from the superhydrophobic surfaces under the influence of electric field. (A) On superhydrophobic FDT surface, the jumping droplets exhibit a continuous upward motion when subjected to an upward electric field, indicating the presence of positive charges in the droplets. (B-C) On superhydrophobic DT surface, the jumping droplets fall

under an upward electric field (B) but continuously ascend under a downward electric field (C), signifying the presence of negative charges. This comparison demonstrates that droplet charges are independent of the electric field.

3.1 Such a surface with needles may also be critical for the Kelvin potential measurements. The surface is certainly not homogeneous and extremely rough. Which part of the surface is dominating the measured Kelvin potential? How does the Kelvin potential vary across the surface?

Response: The Kelvin potential distribution across superhydrophobic surfaces is analogous to that observed on planar surfaces, as demonstrated in Figure R3. Specifically, Figure R3A shows the KPFM image of the superhydrophobic DT surface, along with the variation of the surface potential of the lined area, and Figure R3B shows the same information of highly oriented pyrolytic graphite (HOPG), a standard reference material for KPFM analysis. We find that both the nano-structured DT surface and the HOPG surface exhibit fluctuations in surface potential, suggesting that uniformly distributed nanostructures of superhydrophobic surfaces does not significantly influence the Kelvin potential assessments. The KPFM image of HOPG has been added as Figure S11 in the revised supporting information.

Figure R3. KPFM images and variation of surface potential in selected lined area on superhydrophobic DT surface (A) and on HOPG (B) surfaces.

The surface potential values on superhydrophobic surfaces conform to a normal distribution (Figure R4 or Figure 4B in the main text). Consequently, we defined the values of the most significant intensity (*i.e.*, the peak of the histogram) as the surface potential of surfaces. Further, we conducted tests on the surface potential of three randomly selected regions (each with an area of $2 \times 2 \mu\text{m}^2$) of the superhydrophobic surfaces and then calculated their average values and errors.

Figure R4. Histogram of the surface potential of superhydrophobic surfaces with varied k values. The value corresponding to the most significant intensity (*i.e.*, the peak of the histogram) is defined as the surface potential.

3.2 Is gold a good reference for KPFM? It is notoriously contaminated.

Response: Mant thanks for this professional comment. It's right that the gold surfaces are prone to contamination adsorption. In the revised manuscript, we adopted the highly oriented pyrolytic graphite (HOPG) to serve as a reference for the work functions of binary superhydrophobic surfaces. The HOPG surface was renewed using a scotch tape, and its work function is reported to be 4.4 eV in air

(Surf. Sci., 2001, 481, 1-3, 172-184). Based on the measured surface potential of HOPG (Figure R3B, also Figure S11), we have recalibrated the work function values of superhydrophobic surfaces in Figures 4 and 5 of the revised manuscript.

4. A figure of the setup should be in the main manuscript. Now it is in SI as figure S1. I also suggest adding an SEM image of the Cu nanograss.

Response: Many thanks for this advice. We have repositioned the schematic diagram of the experimental setup and the SEM image of the nano-structured substrate to Figure 2 in the main text, as illustrated in Figure R5.

Figure R5 (also Figure 2 in the main text) CE between coalescence-induced jumping droplets and superhydrophobic surfaces. (A) Molecule structure of FDT and DT, as well as the morphology of nano-structured copper oxide substrate with 12 minutes of etching. The scale bar is 5 μ m. (B) The schematic of the setup that allows the observation of the contact electrification between superhydrophobic surfaces and coalescence-induced jumping droplets. The cooler aims to trigger the condensation of vapor and coalescence-induced jumping of condensate droplets by decreasing the

temperature of superhydrophobic surfaces. Two electrodes build the directional electric field, allowing us to judge the charge polarity of jumping droplets by observing their motion trajectory. (C) On superhydrophobic FDT surfaces, jumping droplets experience a continuous ascent motion under the upward electric field, denoting generated positive charges in droplets during CE. (D) On superhydrophobic DT surfaces, jumping droplets acquire negative charges, leading to initial jumping and followed falling motion under the upward electric field. The contact angles of water droplets on FDT and DT surfaces are $\sim 156.2^\circ$ and $\sim 155.4^\circ$, respectively, shown in the insets.

5. Reporting the density of water with 6 digits precision (note S1) does not make much sense. Especially because the temperature is not reported.

Response: We agree with the reviewer's view about the statement of the water density. Accordingly, we have adopted an empirical value of 998 kg/m^3 for the density of water in the ambient environment in the revised supporting information.

6.1. The authors argue that on hydrophobic surfaces electrons and ions remain on the surface while on superhydrophobic surfaces only electrons remain (e.g. figure 5 caption). What is the evidence? I do not see convincing arguments.

Response: Figure 5A compares charge carriers (electrons and/or ions) during liquid-solid CE occurring on hydrophobic surfaces and on superhydrophobic surfaces. For liquid-solid CE on hydrophobic surfaces, the electron and ion transfer are well-documented in the literature, as substantiated by the references cited in our manuscript (Ref. 18-22), and now explicitly noted in the caption of Figure 5 for clarity.

Here, we specifically reassert the occurrence of electron transfer and the non-occurrence of ion transfer on superhydrophobic surfaces during liquid-solid CE. First, we align with the occurrence of electron transfer during liquid-solid CE documented by prior studies (Ref. 18-23 in the manuscript). However, these studies failed to elucidate how electron transfer is dictated by surface properties. In our work, using the binary superhydrophobic surfaces with adjustable surface potentials, we reveal the linear correlation between generated charges and surface work function, suggesting that electron transfer is driven by the disparity in work function between the liquid-solid surfaces. Importantly, this linear

correlation not only confirms electron transfer but also negates the presence of ions on superhydrophobic surfaces. We then further substantiate the absence of ions by a series of experiments, including characterizing the surface potential and charging ability after contact with ion-enriched liquids, even the liquids with extreme pH values, as demonstrated in Figure 5 and Note S6, Figures S8-10, as well as the response to following comment 6.2.

6.2. The soaking experiments are not convincing to me. It is not clear at all, what the soaking does to the thiol layer on the oxidized copper.

Response: Our understanding is that the reviewer's comment pertains to the objectives of the soaking experiments. Soaking experiments in this work aim to exclude the possibility of ion residues on the superhydrophobic surfaces. Soaking represents one form of contact manners between liquids and solid surfaces, featuring a longer liquid-solid contact time than the droplet impact. Ion transfer has been observed during soaking experiments on hydrophobic surfaces, as indicated by changes in surface potential (Figure R6A, ACS Nano 2020, 14, 12, 17565–17573). However, the surface potentials of superhydrophobic DT and FDT surfaces remain stable when soaked in various aqueous solutions, proving no ion transfer to the superhydrophobic surfaces. Thus, soaking does not alter the surface characteristics of thiol-modified copper oxide. It is also worth noting that the superhydrophobic thiol-modified copper oxide has been widely used in previous work due to its stable surface properties, such as the work by Miljkovic et al. (ACS Nano 2013, 7, 12, 11043–11054).

Figure R6. The influence of soaking on the surface potential of hydrophobic surface (A) and of

superhydrophobic surface. (A) The surface potential of hydrophobic surface (PTFE) changes after soaking with different liquids, reproduced from the work by Zhan et al. (ACS Nano 2020, 14, 12, 17565–17573). (B) The surface potential of superhydrophobic DT and FTD surfaces remains stable after soaking.

Reviewer #3

The authors investigated the mechanisms governing surface-droplet contact electrification using superhydrophobic surfaces that decouple ion transfer and electron transfer. The manuscript was well written and the experiments were well performed. I also like the idea of using binary self-assembled monolayer for a delicate control of droplet charging. In order to further reinforce the strength of the study, the following aspects should be addressed with additional clarity:

Response: We deeply appreciate your recognition of our idea, and we sincerely thank you for your valuable feedback. All the comments are very valuable for us to improve this work, and they have been carefully addressed in the revised manuscript. We sincerely hope that the revised version will meet your rigorous criteria for publication.

1. The title and Figure 1 appear to emphasize the unique mechanism of contact electrification associated with superhydrophobic surfaces in comparison to hydrophilic and hydrophobic surfaces. Based on my understanding, superhydrophobic surfaces can be regarded as an extension of the hydrophobic case when considering droplet-surface interactions at a microscale level. It remains unclear what fundamentally differentiates droplet-contact induced electrification on smooth hydrophobic surfaces from superhydrophobic surfaces. If the distinction lies in the presence of ion residue, it would be valuable to discuss the implications of using a slippery surface.

Response: We understand the reviewers' concerns about our seemingly counterintuitive conclusions on the liquid-solid contact electrification (CE) occurring on superhydrophobic in comparison to that on hydrophobic surfaces. Wettability has never been a main character in previous liquid-solid CE studies in which hydrophobic surfaces and hydrophilic surfaces share the same triboelectric mechanism. If there are also no differences between hydrophobic and superhydrophobic surfaces, it would imply that the same situation for liquid-solid CE occurring on superhydrophobic surfaces and hydrophilic surfaces, appearing to be more contradictory to common sense.

The most apparent distinction between superhydrophobic surface and hydrophobic surface lies in their interaction with the liquids, which determines whether ion transfer/residue occurs during liquid-solid CE. On hydrophobic surfaces, contact with the water inevitably leads to liquid residues on surfaces,

as these polymeric surfaces, such as FEP and PTFE, can even adsorb water from the air (Angew. Chem. Int. Ed. 2008, 47, 2188-2207). A recent study also found that liquid dewetting from the hydrophobic surface (coated with trichloro(1H,1H,2H,2H-perfluorooctyl) silane) leaves behind a hydration shell, further confirming the presence of liquid residue during contact (arXiv preprint arXiv:2305.02172, 2023). During such a process, ions in liquid are also deposited on the surfaces, analogous to the situation that the evaporation of a sodium chloride solution on surfaces usually leads to the precipitation of sodium chloride crystals. However, such a source of transferred ions has never been recognized by previous liquid-solid CE studies. In this work, we introduce the superhydrophobic surface to decouple the ions and electrons, two kinds of charge carriers, during liquid-solid CE by excluding the process of liquid residue and ion residue on the surfaces, thereby achieving an in-depth understanding of liquid-solid CE.

We further outlined ion-transfer-induced differences between liquid-solid CE occurring on the hydrophobic surfaces and superhydrophobic surfaces in Table R1. During contact with the liquid, the hydrophobic surfaces gradually adsorb ions (ACS Nano 2020, 14, 12, 17565–17573; Adv. Mater. 2020, 32, 1905696; Nat. Commun. 2020, 11, 399). These ions alter the surface potential (or work function) of hydrophobic surfaces, which diminishes the original driving force of liquid-solid CE, thereby suppressing the following charge generation, *i.e.*, degradation of the charging ability. In contrast, superhydrophobic surfaces that are impacted or soaked by the water and ion-enriched liquids, including acid, alkali, and salt aqueous solution, maintain a stable surface potential, which simultaneously means a stable charging ability, as discussed in Figure 5 of the manuscript and Note S5 in supporting information. As a result, the liquid-solid CE occurring on the superhydrophobic surfaces demonstrates independence on the pH values of the liquid. In other words, superhydrophobic surfaces can either negatively or positively electrify the liquids regardless of pH values. In contrast, liquid-solid CE occurring on hydrophobic surfaces is usually dependent on liquid pH, which manifests that the charge polarity is determined by the pH of the liquid and the isoelectric point of the solid surfaces due to the competitive adsorption of cations and anions on solid surfaces, as discussed in Note S6 in supporting information. Furthermore, the stable surface potential of superhydrophobic surfaces allows us to find the linear correlation between surface potential and generated charges, thereby offering us a pave for quantitatively controlling the charge magnitude, which cannot be achieved on

hydrophobic surfaces with dynamically changing surface potential during CE. These newly found experimental results and underlying mechanisms imply the difference between liquid-solid CE occurring on superhydrophobic surfaces and hydrophobic surfaces are not limited in wettability.

	Hydrophobic (Previously studied)	Superhydrophobic (Currently focused)
Adsorbed ions	Increase until saturation	No ions
Stable surface potential	×	√
Stable charging ability	×	√
pH independence	×	√
Quantitatively control of charges	×	√
↓		
Charge carriers	Electrons and ions	Electrons

Table R1. Comparison of liquid-solid CE occurring on hydrophobic surfaces (previously studied) and on superhydrophobic surfaces (currently studied).

2. The claim that superhydrophobic surfaces eliminate liquid and ion residues should be approached with greater rigor. From a microscopic perspective, the droplet interacts with the surface structures, and it is plausible that residues persist on the micro/nanoroughness after liquid-droplet separation. This is especially relevant when the droplet possesses a higher temperature than the substrate, as condensation underneath the droplet (for instance, ACS nano, 13(7), 8169-8184, 2019) during impact may result in the retention of residues on the surface.

Response: We strongly agree with your view about the rigor statements for the elimination of liquid and ion residues on the surface, especially after carefully referring to the reference recommended and other relevant sources.

Based on the calculations described in the work (ACS nano, 13(7), 8169-8184, 2019), we quantify the size of droplets possibly situated in in nanostructures during our droplet impact or soaking experiments.

Considering contact time scales of 10 milliseconds and 1 minute in these two experiments, the droplet volumes are calculated to at the order of magnitude of 10^{-4} and 10^{-1} nL, respectively. These droplets possibly are too small or fall below the detection limit to show an obvious effect on our results in detecting ions on surfaces. However, as the reviewer astutely points out, condensation becomes particularly relevant when the droplet possesses a higher temperature than the substrate, such as in the situation of liquid flooding on the surface, which may bring a different result in liquid-solid CE. Similarly, a hydrostatic pressure that is large enough to cause the Cassie-to-Wenzel transition of droplets on superhydrophobic surfaces (Appl. Phys. Rev. 2021, 8, 031403) will also be accompanied by the liquid and ion residues on the surfaces. In light of these considerations, we have enhanced the manuscript by providing a more robust description of these possibilities to cause the failure of superhydrophobic surfaces, as below “Note that failure of superhydrophobic surfaces under certain conditions, such as overloaded hydrostatic pressure or liquid flooding, may lead to the occurrence of liquid residues induced ion transfer during CE.”

3. A more comprehensive characterization of the binary self-assembled monolayers employed to achieve superhydrophobicity is suggested, particularly with regard to the uniformity and thickness of the coating. In addition to chemical adsorption, physical adsorption can occur in certain cases, potentially resulting in the deposition of additional layers of molecules onto the monolayer. This phenomenon may surpass the controllability achieved solely by manipulating the molar ratio of molecules.

Response: Many thanks for the professional suggestion. It’s right that uniformity and thickness are crucial for the self-assembled monolayers. Over the past two decades, both single-component and binary mercaptan-based self-assembled monolayers have been widely used to tune the work function of the surfaces in the optoelectronic field (Adv. Mater. 2005, 17, 621; Langmuir 2009, 25, 11, 6232–6238; J. Phys. Chem. B 2003, 107, 11690-11699). These studies have revealed that self-assembled monolayer constitutes a mature coating technology with well-controlled uniformity. However, our work did not specifically address these features of binary self-assembled monolayers. Accordingly, we made the response below and added the supplement to the revised work.

To acquire the uniform self-assembled monolayers, we implemented a careful rinsing process on the

dip-coated surfaces using alcohol, followed by drying with nitrogen blow. This step was crucial in dramatically reducing the extent of physical adsorption, which has been supplemented in the experimental part of the revised manuscript (Line 259 Page11). Without the rinsing step, directly drying the surface would result in a non-uniform appearance, indicating intensive physical adsorption. To further demonstrate the uniformity of binary superhydrophobic surfaces, we analyzed their surface potential (Figure R7, also included in Figure S5 in the revised supporting information). The surface potentials of DT and DT range from 700 to 880 mV and -1200 to -1300 mV, respectively. The binary surface with k values (*i.e.*, the molar ratio of FDT) of 0.2 ranges from -500 to -700, signifying the cooperative contribution of FDT and DT to the surface potential. If binary surfaces are non-uniform, with a clear separation between FDT and DT regions, the surface potential would span from 880 to -1300 mV. Furthermore, the measured surface potential values (or work functions) across any randomly selected regions on binary superhydrophobic surfaces possess a small variance (Figure 4F), also speaking for surface uniformity.

Figure R7. KPFM images of superhydrophobic surfaces with varied k values, denoting the uniformity of surfaces.

Regarding the measurement of film thickness, we apologize for being unable to find a suitable method. This is because ellipsometry, which is a widely used tool for characterizing the thickness of nm-scale self-assembled monolayers, fails in our work due to the light-absorption property of the nano-structured copper oxide used. However, we believe that the surface uniformity, which was discussed above, can alleviate the concerns of reviewers regarding the impact of physical adsorption on the control of the molar ratio on the superhydrophobic surfaces.

4. Although the driving force (work function) has been identified, the impact of droplet-surface interaction on electron transfer remains to be further illuminated. The authors have demonstrated that the obtained charges are also influenced by the surface morphologies of the superhydrophobic surfaces. This suggests that droplet impact dynamics may affect the actual contact area between the droplet and the surface, consequently influencing the quantity of charges acquired. Bridging the gap between droplet charges and droplet-surface interaction would significantly enhance the current study.

Response: In response to this comment, we elucidate the correlations between droplet impact dynamics and droplet charges. In our experiments for quantifying the generated charges during CE, the interactions between droplet and superhydrophobic surface are determined by the tilt angle and dropping heights (figure R8A) due to their influence on the maximum spreading area (A_{max} in Figure R8B) and sliding length (Figure R8C), which further affects the generated charges. Figure R8D illustrates that A_{max} (left y-axis) decreases while the sliding length (right a-axis) increases with the increase of tilt angles of the superhydrophobic surfaces. We find that maximum charge generation in water at the tilt angles of 45° (Figure R8E), with neither the largest A_{max} nor largest sliding length, indicating the coefficient of A_{max} and sliding length on charge generation. Regarding the effect of dropping height on charge generation, Figure R8F demonstrates that the magnitude of generated charges gradually increases and then saturates with the increase of dropping height. In a large dropping height (for example, 4.5 cm), the droplet bouncing off the superhydrophobic surface typically fragments into two daughter droplets, during which new types of charges (may not be triboelectric charges) may be generated and thus, affect the quantification of triboelectric charges. Accordingly, the largest dropping height we employed is 4 cm. We have included this information in Note S4 and Figure S7 of the revised supporting information.

Figure R8 (also see Figure S7). Influence of droplet impact dynamic on charge generation during CE between water and superhydrophobic surfaces. (A) Schematic diagram illustrating the parameters that affect droplet impact dynamic. (B) Optical images of the water droplet impact on the tilted superhydrophobic surfaces. A_{max} represents the maximum spreading area of the water droplet. (C) A time-lapsed droplet trajectory that records the sliding process of droplets after impact on the surfaces. (D) The A_{max} and sliding length of impacted droplets on the superhydrophobic surfaces with varied tilt angles. The data are derived from figures and videos that were analyzed using the ImageJ software. (E) Droplet charges generated during CE between water and superhydrophobic FDT surfaces with varied tilt angles. The dropping height in (B-E) is 3 cm. (F) The variation of charges generated in water as the function of dropping height. The magnitude of generated charges gradually saturates with the increase of dropping height. The error bars in (D-F) are based on the SD values of three tests.

5. The authors primarily quantified the charge of the droplet rather than that of the surface. Therefore, "How do the superhydrophobic surfaces charge the liquid droplet?" seems a more suitable title than the current version, "How does the liquid charge the superhydrophobic surface?"

Response: We are thankful for the reviewer's suggestion pertaining to the title of this work. Scientifically, the generated charges on liquid and superhydrophobic surfaces during contact electrification feature equal magnitude but opposite polarity; hence, droplet-charging superhydrophobic surface and superhydrophobic surface-charging droplets are equivalent. Technically,

quantifying droplet charges is more straightforward and reliable than quantifying the charges on superhydrophobic surfaces for the following reasons. The charges in a single droplet are too few to distinguish the charging abilities among the superhydrophobic surfaces with varied k values. Accordingly, total charges in more droplets should be recorded, which is easily achieved, as shown in our main text. However, the accumulation of charges on superhydrophobic surfaces should be avoided because these charges could alter the surface potential (or work function) and inhibit the following charge generation. Hence, we grounded the superhydrophobic surfaces to prevent charge accumulation on the surfaces and measured the total charges of multiple droplets. Considering the reasons above, we prefer the current version of the title.

6. Lastly, the following typos should be corrected:

- "Self-assembly monolayer" should be written as "self-assembled monolayer."

- In Supplementary Note S1, the sign of the gravitational force mg should be positive.

Response: Thank the reviewer for carefully examining our works and pointing out our typos. We have corrected these typos and made further checks of the revised manuscript.

REVIEWER COMMENTS

Reviewer #1 (Remarks to the Author):

I am overall very happy with the review responses that have been made by Wang and co-workers. I think the manuscript is in much better shape than it was before. There are however, still some issues that should be addressed further by the authors before it is finally ready to be accepted for publication

1) The first issue is in relation to the usage of GGA functionals for their calculations. While I fully understand the point that they have made in the the response, I insist that the authors need to perform some benchmarks with more accurate functionals like hybrids in order to validate their results.

The U correction essentially corrects for the self-interaction error which exists in any system - water is a large-band gap insulator but without taking into account the U or Hybrids, it tends to overdelocalize charge and lower the band gap. I therefore would like the authors to perform some tests with hybrids in order to validate that their main conclusions are robust.

2) The initial reference that I had included for the comment on the OH- was just an initial point of reference that I had hoped would expand the discussion around the propensity of protons, hydroxide ions and even surface impurities. I think the authors should expand their text to cover ALL these possibilities. In particular, they should look at all the following papers by different groups and include them into their manuscript:

OH- at water-vacuum interfaces:

<https://www.sciencedirect.com/science/article/pii/S0009261409010938>

<https://doi.org/10.1021/jp906978v>

H+ and OH- at water vacuum interfaces:

<https://pubs.acs.org/doi/10.1021/jp501854h>

<https://doi.org/10.1063/1.4941107>

Surface Impurities:

<https://doi.org/10.1016/j.coelec.2018.09.003>

<https://doi.org/10.1063/5.0127869>

The last set of literature on Impurities is really important to get into more deeply. How the authors eliminate that the charging effects they see are not due to trace impurities?

A comment/discussion on this highlighting the literature above is warranted.

Reviewer #2 (Remarks to the Author):

The authors have made substantial changes to the manuscript. The experiments are interesting and the work should be made available to the public. However, I still think that it is relevant only for a

narrow community because the specific way of drop charging is very special. It is from a metal (Cu) to water drops under the influence of strong electric fields. The conductor is highly structured so that at the sharp edges very high electric fields can be generated. Electrons can easily be ejected or taken up under such conditions. This electron transfer could be modified by the presence of the different thiols. I do not see how conclusion with respect to the usually charging mechanism of drops on hydrophobic insulating surfaces can be drawn. For this reason, I am still not convinced that Nature Communications is the right journal.

Additional comment:

The first sentence is: "Liquid-solid contact electrification (CE) is essential to diverse applications, such as water energy harvesting". This statement is currently not correct. To my knowledge, there is no electricity produced by liquid contact electrification. Even the most optimistic predictions expect niche applications.

In the 4th sentence of the abstract, the authors say that "An emerging view is that both electrons and ions serve as charge carriers, in which electron transfer dominates while ion transfer plays a subsidiary role". My perception of the literature is different. I do not see the dominant role of electrons.

In the introduction the charging mechanism between a solid and liquid is described as one universal phenomenon called "contact electrification". I find this misleading. Depending on the materials, their topology and the specific process, different mechanisms may dominate. The description does not take into account, that charge separation may depend on the process and may be fundamentally different for different materials.

Reviewer #3 (Remarks to the Author):

After going through the revised manuscript, I believe the authors have addressed my concerns. I thus support the publication of this work.

—Xiao Yan

Reviewer #1

I am overall very happy with the review responses that have been made by Wang and co-workers. I think the manuscript is in much better shape than it was before. There are however, still some issues that should be addressed further by the authors before it is finally ready to be accepted for publication.

Response: We appreciate the reviewer for the positive feedback and insightful comments. We have made the corresponding revisions to improve our manuscript further. We sincerely hope that the revised manuscript will satisfy your stringent criteria for publication.

1. The first issue is in relation to the usage of GGA functionals for their calculations. While I fully understand the point that they have made in the response, I insist that the authors need to perform some benchmarks with more accurate functionals like hybrids in order to validate their results. The U correction essentially corrects for the self-interaction error which exists in any system - water is a large-band gap insulator but without taking into account the U or Hybrids, it tends to over delocalize charge and lower the band gap. I therefore would like the authors to perform some tests with hybrids in order to validate that their main conclusions are robust.

Response: Thanks very much for this kind suggestion. Based on this comment, we have carried out the calculations of the DT and FDT based on both hybrid functionals and GGA+U as shown in **Figure R1**. For hybrid functionals, we have applied the HSE06 functionals with norm-conserving pseudopotentials and a cutoff energy of 630 eV. The k-point setting has been set to ultrafine quality and the convergence criteria have remained the same. For the GGA+U functional, the numbers of valence states have been treated as (2s, 2p), (2s,2p), (3s,3p), and (1s) for C, F, S, and H, respectively. It is noted that compared to the original calculations with GGA+PBE functionals, the overall electronic structures have not been significantly affected by the functionals or U corrections.

Compared to Figure R1a and R1d, we have noticed that the overall electronic structures basically remain similar. The bandgap has only been minorly increased in hybrid functionals and DFT+U methods and more splitting peaks appeared in the p orbitals, which also clearly reveals the p-p orbital coupling among C, F, and S. We have noticed that the S-3p orbitals have been upshifted, which are located on the Fermi level for both hybrid functionals and DFT+U methods. Although the conduction band minimum (CBM) has been slightly upshifted, the bandgap of FDT still remains slightly larger

than DT, which is consistent with our conclusions. It is worth noting that F-2p orbitals become electron-rich obtained by the hybrid functionals and DFT+U method, leading to the downshift of the overall p-band center of FDT. Based on the current results obtained by the hybrid functionals and DFT+U methods, we have confirmed that the conclusions of our work will not be affected by the selection of the GGA+PBE functionals. According to the Reviewer's suggestion, we have updated the PDOS of FDT and DT in Figure 3 C-E with the hybrid functional calculation results, which do not affect our previous conclusions and descriptions of Figure 3. The corresponding calculation details have also been updated.

Figure R1. The PDOS calculation of FDT by (a) GGA+PBE, (b) hybrid functional-HSE06, and (c) DFT+U methods. The PDOS calculation of DT by (c) GGA+PBE, (d) hybrid functional-HSE06, and (e) DFT+U methods.

2. The initial reference that I had included for the comment on the OH⁻ was just an initial point of reference that I had hoped would expand the discussion around the propensity of protons, hydroxide ions and even surface impurities. I think the authors should expand their text to cover ALL these possibilities. In particular, they should look at all the following papers by different groups and include them into their manuscript:

OH⁻ at water-vacuum interfaces:

<https://www.sciencedirect.com/science/article/pii/S0009261409010938>

<https://doi.org/10.1021/jp906978v>

H⁺ and OH⁻ at water vacuum interfaces:

<https://pubs.acs.org/doi/10.1021/jp501854h>

<https://doi.org/10.1063/1.4941107>

Surface Impurities:

<https://doi.org/10.1016/j.coelec.2018.09.003>

<https://doi.org/10.1063/5.0127869>

The last set of literature on Impurities is really important to get into more deeply. How the authors eliminate that the charging effects they see are not due to trace impurities?

A comment/discussion on this highlighting the literature above is warranted.

Response: We deeply feel the reviewer's efforts to help us strengthen this work. Thanks very much.

In this response, we will separately discuss our revision from the aspect of the ions and impurities.

We acknowledge the significance of discussing the propensity of cations and anions on the hydrophobic surfaces in explaining the formation of charged liquid/hydrophobic interfaces. In the *Introduction* section of the revised manuscript (Line 60, Page 3), we emphasize that the controversial ion-transfer model results not only from the debatable surface affinity of anions but also from that of cations (Ref. 32-36). Motivated by these literatures, we have also included additional information about the dependence of ion propensities on the liquid pH in the *Introduction* section. However, we have refrained from more discussion about the ion-transfer model or ion propensities, given that ion transfer is not involved in CE between liquid and superhydrophobic surfaces, which otherwise divert our focus and potentially overwhelm the readers with excessive information.

Regarding surface impurities, we have included additional discussion to exclude their influence on our results in the revised manuscript (Line 210, Page 9). Surface impurities are usually introduced to account for some phenomena beyond ion adsorption, and their identities remain poorly understood at present. However, it is a well-established fact that surface impurities, behaving similarly to cations and anions, also induce the pH-dependent feature of the polarity (surface potential or zeta potential) on the

charged surfaces (Langmuir 2020, 36, 3645–3658). In contrast, the CE between liquid and superhydrophobic surfaces shows a unique property of pH independence, suggesting the absence of impurities on the superhydrophobic surfaces. Note that in the revised manuscript, we have cited the fifth reference recommended by the reviewer, as well as another study conducted by the same research group (Ref. 46 and 47).

Reviewer #2

The authors have made substantial changes to the manuscript. The experiments are interesting and the work should be made available to the public. However, I still think that it is relevant only for a narrow community because the specific way of drop charging is very special. It is from a metal (Cu) to water drops under the influence of strong electric fields. The conductor is highly structured so that at the sharp edges very high electric fields can be generated. Electrons can easily be ejected or taken up under such conditions. This electron transfer could be modified by the presence of the different thiols. I do not see how conclusion with respect to the usually charging mechanism of drops on hydrophobic insulting surfaces can be drawn. For this reason, I am still not convinced that Nature Communications is the right journal.

Response: We greatly appreciate the reviewer's recognition of our revised work, and we realize that our improvements owe much to the reviewers' critical comments and professional suggestions.

The reviewer holds the belief that our droplet charging method is unique due to the utilization of a strong electric field, leading to the rejection of our work for publication in Nature Communications. However, we believe that there is a misunderstanding regarding the nature of our study, and we would like to provide an explanation.

First, it is important to note that the liquid-solid contact electrification is irrelevant to the application of the electric field. To provide a clear explanation, we have added the subheading in the revised manuscript. In the section titled "CE between water droplets and binary superhydrophobic surfaces with regulable work functions" and the subsequent texts, there is no application of the electric field. The liquids acquire static charges upon impacting and rebounding from the grounded superhydrophobic surfaces, and the charge polarity and magnitude are dependent on the k values of superhydrophobic surfaces. This demonstrates our successful manipulation for the liquid-solid contact electrification and electron transfer through the use of the binary superhydrophobic surfaces, rather than through the electric field.

Second, the electric field is only applied in the section titled "CE between coalescence-induced jumping droplets and superhydrophobic surface", aiming to visually assess the charge polarity of jumping droplets based on their motion trace. In such a scenario, the electric field cannot exert any

influence on the charge generation and charge polarity during contact electrification.

Additional comment:

1. The first sentence is: “Liquid-solid contact electrification (CE) is essential to diverse applications, such as water energy harvesting”. This statement is currently not correct. To my knowledge, there is no electricity produced by liquid contact electrification. Even the most optimist predictions expect niche applications.

Response: We have removed the statement “such as water energy harvesting” from the abstract of the revised manuscript.

However, we respectfully disagree with the reviewer's assertion that there is no electricity produced by liquid contact electrification. In recent years, liquid-solid contact electrification has been extensively studied and reported by global academic communities in the field of water and droplet energy harvesting. For example, our research group reported a droplet-based electricity generator that harvests the energy generated during the liquid-solid contact electrification (Figure R2, Nature, 2020, 578, 392-396, also see Ref. 9 in manuscript). Additionally, we would like to highlight several notable works in this area, including those by Prof. Frieder Mugele from the University of Twente (Adv. Mater., 2020, 2001699; Phys. Rev. Lett. 125, 078301), Prof. Zhong Lin Wang from the Chinese Academy of Sciences (Angew. Chem., 2013, 125, 12777 –12781, Adv Mater., 2014; 26, 4690 - 4696.), and Prof. Hyuk Kyu Pak from Pusan National University (Nat. Commun. 2013; 4,1487). By providing such information, we aim to emphasize the key role of liquid-solid contact electrification in generating electricity, as supported by numerous research studies. The harvested electric energy has also been applied in diversified applications, such as chemical sensing, microfluidic actuation, and so on (Chem. Rev., 2022, 122, 5209–5232).

Figure R2. Configuration of a typical droplet-based electricity generator that can harvest the energy

from liquid-solid contact electrification. The generated electricity can light hundreds of LEDs. The figures are from the Nature, 2020, 578, 392-396.

2. In the 4th sentence of the abstract, the authors say that “An emerging view is that both electrons and ions serve as charge carriers, in which electron transfer dominates while ion transfer plays a subsidiary role”. My perception of the literature is different. I do not see the dominant role of electrons.

Response: To avoid any possible misinterpretation, we have removed such a statement that may not be a universally accepted consensus.

However, we would still like to engage in a discussion with the reviewer. The assertion regarding the dominant role of electrons during liquid-solid CE is not a subjective viewpoint we put forth, but rather a summary of the existing studies (Ref. 18-22 in the revised manuscript). Dating back to a decade ago, the dominant charge carriers, recognized by the communities, are ions, as we discussed in the *Introduction* section. Such a transition from ions to electrons as the primary charge carriers represents the progressive advancement in comprehension of liquid-solid contact electrification. In our study, we contribute further to this field by exploring the nature of charge carriers and how their selection is influenced by surface properties.

3. In the introduction the charging mechanism between a solid and liquid is described as one universal phenomenon called “contact electrification”. I find this misleading. Depending on the materials, their topology and the specific process, different mechanisms may dominate. The description does not take into account, that charge separation may depend on the process and may be fundamentally different for different materials.

Response: To be honest, we cannot fully grasp the reviewer’s concerns because we have thoroughly reviewed the various liquid-solid charging manners in the manuscript, including contact electrification, introduction electrification, conduction electrification, and charge transfer across the hydrogen bonds, as explicitly discussed in the *Introduction* section. Our statement in the manuscript is shown below: “It is worth mentioning that charge generation in water or at water-involved interfaces can arise not only from the widely known CE, but also from the other various manners, including introduction and conduction electrification, as well as the charge transfer across the hydrogen bonds.”

If the reviewer's concern pertains to the content in Figure 1, we would like to clarify that all the solid materials depicted in Figure 1 have been reported in the studies related to liquid-solid contact electrification. We just classify the contact electrification between these materials and liquids based on the considerations of surface wettability and the charge polarity of water.

If we have misinterpreted the reviewer's concerns, we would like to discuss it further with the reviewer.

Reviewer #3

After going through the revised manuscript, I believe the authors have addressed my concerns. I thus support the publication of this work.

—Xiao Yan

Response: We appreciate the reviewer's recommendation of publication.

REVIEWERS' COMMENTS

Reviewer #1 (Remarks to the Author):

I am happy with all the revisions made by the reviewers and now the paper can be accepted for publication.